# Symmetry-Breaking Stabilities in Carapace Curvature on *Testudo* (Reptilia, Testudinidae)

**DOI:** 10.3390/ani12040471

**Published:** 2022-02-14

**Authors:** Pere M. Parés-Casanova, Joaquim Soler, Tania Buisán, Albert Martínez-Silvestre

**Affiliations:** 1Escola Agrària del Pirineu, Finca Les Colomines (Bellestar), 25711 Montferrer i Castellbò, Catalonia, Spain; 2CRARC, Catalonia Reptiles and Amphibians Rescue Center, 08783 Masquefa, Catalonia, Spain; crarc@amasquefa.com (J.S.); albertmarsil@hotmail.com (A.M.-S.); 3Independent Researcher, 22422 Fonz, Huesca, Spain; taniabar3a@gmail.com

**Keywords:** developmental instability, directional asymmetry, morphometrics, shell, Testudinata, turtles

## Abstract

**Simple Summary:**

When turned upside down, terrestrial turtles have no active control on self-righting if the animal has been flipped over. Turtles belonging to *Testudo* genus present high-domed carapaces and short neck and legs. Using geometric morphometric techniques, we studied left-right frontal symmetry among some *Testudo* species to study if carapaces’ geometry may serve as the tool to roll right side up. A clear right directionality was detected in the studied sample. This fact. more that easing the self-righting potential (“kinematic instability”, understood as the ability to self-right without effort), would make the stable ventral turning difficult (“static stability”, understood as the ability to resist passively turning of the body produced by destabilizing forces).

**Abstract:**

The aim of this research was to contribute to the study of the doming geometry of *Testudo* carapace as an unstable point of equilibrium when animals are overturned. We performed this research using geometric morphometric using a sample of 64 *Testudo* individuals belonging to different species (*T. hermanni*
*n* = 30, *T. graeca*
*n* = 3, *T. marginata* *n* = 13 and *T. horsfieldii* *n* = 18), sexes and ages. A set of four sagittal landmarks (discrete homologous points) and 15 pairs of semi-landmarks, on the frontal doming of the carapace, were digitized on individual carapace pictures. Significative fluctuating asymmetry was detected, defined as small, completely random departures from bilateral symmetry, but much less than directional asymmetry, which appeared highly significative. Anti-symmetry did not appear. Carapace asymmetry was dominated by a clear right directionality. A possible biological speculation could be that this asymmetry more that easing the self-righting potential (“kinematic instability”, understood as the ability to self-right without effort), makes stable ventral turning difficult (“static stability”, understood as the ability to resist passively turning the body produced by destabilizing forces). This asymmetry is present among both sexes but more marked among males. An explanation for this sexually differentiated pattern could be the higher locomotion and the fight for mating in males, making them consequently more prone to losing their balance and falling on their back. These data may be useful in studying adaptative traits in *Testudo* species as well as establishing a seminal base for future studies. This research is the first attempt to explore a suitable method to assess doming asymmetry which could be useful in future, more extensive investigations, on a larger interspecific sample.

## 1. Introduction

Turtles have an anatomy dominated by the shell, a structure derived from elements of the axial skeleton [1]. A shell is composed of the carapace (dorsal elements), and the plastron (ventral plate), enclosing the locomotor elements in a situation unique among tetrapods [1].

When turned upside down, terrestrial turtles can self-right [2] but with their short limbs and neck, they have no active control on self-righting if the animal has been flipped over. Here, a carapace’s geometry may serve as the tool to roll right-side up [2,3]. This must be especially important for species that have high-domed carapaces and short legs. If a domed carapace is not dorsally symmetrical, it seems that this useful instability would help animals in self-righting, so asymmetrical dome-backed tortoises practically would roll back into position almost passively. Much of the literature has been devoted to documenting specific variations of shell [4,5,6,7,8], but there is a scarcity of studies that explicitly test the carapace form from this functional point of view [9,10].

Object asymmetry considers that while the axis of symmetry passes through the entire structure, the right and left sides are not totally flipped mirror images of each other [11]. The geometric dorsal asymmetry of carapaces among high-domed turtles could guarantee instable equilibriums and in consequence, easiness in self-righting. Viscerae probably contributes to this instability too. For instance, the stomach is always to the left of the turtle. If the stomach is full, it will affect the point of gravity by moving it to the left. In addition, the right hepatic lobe is bigger than the left, causing displacement of weight towards the right. In addition, an ovulating female will have an asymmetric imbalance of mass in function of where she has more full follicles [12]. In conclusion, an asymmetric carapace but also internal viscerae can affect the body’s stability point, e.g., the center of mass cannot be assumed to be in the geometric center of the shell.

*Testudo* is a genus of largely arid-adapted tortoises [13]. The most recent and most thorough revision identifies five species: *T. hermanni*, *T. graeca*, *T. marginata, T. kleinmani* and *T. horsfieldii* [14]. Here, we study specifically whether the geometry of highly domed terrestrial species *Testudo* (Linnaeus, 1758)—*T. hermanni*, *T. graeca* and *T. marginata*—acts monostatically, representing an unstable point of equilibrium when animal is overturned, so it can self-right with low effort. We perform this research using geometric morphometric (GM) techniques, which allows the assessment of shape asymmetries.

## 2. Materials and Methods

### 2.1. Sample

A sample of 64 *Testudo* individuals (corpses) belonging to different species (*T. hermanni*
*n* = 30, 15 males and 15 females, *T. graeca*
*n* = 3, 1 male and 2 females, *T. marginata n* = 13, 4 males and 9 females and *T. horsfieldii n* = 18, 10 males and 7 females, and 1 of unknown sex) was studied by means of geometric morphometric techniques. Individuals were obtained from CRARC, Catalonia Reptiles and Amphibians Rescue Center. Housing conditions, temperature regulation, food and water supply for all tortoises corresponded to different practices. All animals were adults of different ages. No carapace showed signs of injury or accessory scutes. Although the sex of turtles influences the overall shape of the carapace in a subtle manner, sex was initially considered.

### 2.2. Imaging

Image captures were performed with a Nikon^®^ D70 digital camera (Nikon Corp., Tokyo, Japan; image resolution of 2240 × 1488 pixels) equipped with a Nikon AF Nikkor^®^ 28–200 mm telephoto lens and JPG file format was used. A tripod with the camera was levelled vertically and the camera was well centered in parallel to the frontal aspect of each carapace, which lied horizontally on its plastron.

### 2.3. Geometric Morphometrics

Pictures were exported to TPSUtil v. 1.70 [15] and then digitized utilizing TPSDig2 v. 1.40 [15]. A set of 4 sagittal landmarks (discrete homologous points) and 15 semilandmarks (equidistant points on an outline determined by extrinsic criteria) per side on the frontal doming of the carapace was digitized on each individual carapace (Figure 1). Sagittal landmarks were used to define the axial mirror plane. The semi-landmarks number was considered enough to capture the curvature of the shell.

The semi-landmarks were important for quantifying shape in areas of the carapace sides that lack clear homologous points. Digitalization was bi-replicated to establish the measurement error. The semi-landmarks were ulteriorly slid using bending energy with TPSUtil v. 1.70 [15]. A generalized least square fitting was performed on all landmark configurations in order to translate (according to the configuration’s centroid), optimally rotate, and adjust for size [16]. Size was computed as centroid size (CS), the square root of the sum of squared distances from each landmark to the specimen’s centroid [17]. A consensus (mean) configuration was obtained. Resulting from this procedure are pure shape data (Procrustes coordinates) and centroid size, which is an important measure for the size of the animal when other data such as body size or weight are not available. To detect outliers, we considered extreme values those which were more than three times the interquartile range.

Fluctuating Asymmetry (FA), Directional Asymmetry (DA) and Antisymmetry (AS) were tested. To detect FA and DA, a Procrustes ANOVA was used. In this analysis, the individuals effect denoted the individual variations of shape and size of each animal, the individuals mean square was the measure of total phenotypic variation, the main effect of *sides* indicated the variation between sides, this indicating DA, and the individuals × sides was the mixed effect and considered as the measure of FA [18]. Measurement error represented the variation due to measurement error in taking landmarks of the same individual in separate sessions [18]. FA refers to small random deviations from perfect bilateral symmetry [19,20]. DA is expressed normally as a greater development of a focal trait on one side of the plane [21,22]. A mean square of DA and FA effect higher than measurement error indicates that the digitizing error is lower than the shape variation among individuals [19]. In Procrustes ANOVA, there are more degrees of freedom than in conventional ANOVA, because the squared deviations are summed over all the landmark coordinates (instead of a single sum of squares in conventional ANOVA) [8]. Therefore, the number of degrees of freedom is that for ordinary ANOVA multiplied by the shape dimension, which is, for our two-dimensional coordinate data, twice the number of landmarks minus four (the number of coordinates minus two dimensions for translation and one each for scaling and rotation) [18]. The ANOVA was performed with 1000 permutations. MANOVA was used for determining the significance of DA (‘side’) of asymmetry component shape variation (parametric) in a covariate matrix using the Pillai’s trace criterion of asymmetry component shape variation. To detect AS, we performed an analysis of the scatter plot of differences between right and right sides and a pairwise correlation of PLS score between both sides [23].

When studying morphological variation, estimation of the allometric effect (defined as the dependence of shape on size) is a necessary step. We investigated by performing a regression of Procrustes coordinates on CS as independent variable. CS was transformed to its logarithm to increase the fit of the model. The number of randomization rounds was 10,000. Additional analysis was performed on regression scores. Canonical Variate Analysis (CVA) was used to ordinate specimens to maximize separation of species with respect to their within-group variances. The degree of asymmetry between sexes was studied by means of a CVA too. To detect overall patterns of shape asymmetrical variation, we performed a Principal Component Analysis (PCA) with regression residuals. The obtained grid is a visual aid and helped to improve the visibility of subtle morphological changes.

For all statistical analyses, we used MorphoJ software v. 1.07a [24] and PAST software v. 2.17c [25], with α = 0.05.

## 3. Results

No outlier was detected. Procrustes ANOVA showed that the variance due to landmark digitization was lower than the variance explained by the shape differences between individuals (individual mean squares = 0.000223744; error mean squares = 0.000188097) (Table 1) which makes the measurement error to be discarded as a confounding factor. As the variation among individuals was higher than that by the digitizing procedure, obtained data can be considered to indicate real biological differences. The MANOVA, which circumvents the assumption of isotropic variation, gave similar results for DA (Pillai trace = 0.86; *p* < 0.0001) but then rendered a significative FA (Pillai trace = 17.42; *p* < 0.0001).

Shape variation also included the effect of allometry. Multivariate regression of Procrustes coordinates onto log centroid size showed that the allometric effect was not significant (*p* = 0.818, based on a permutation 10,000 rounds) and that size accounted only for 0.43% of asymmetrical shape variation. On CVA, the asymmetric shape differences in each species group largely overlapped on the first two canonical variate axes (Figure 2. The exploratory PCA of shape asymmetric component regression scores showed that there was a clear lateral right displacement, with flared marginal scutes (Figure 3). The first three principal components (PCs) did explain a 60.7% of the total variation observed (PC1 + PC2 + PC3 = 26.94% + 21.76% + 12.02% = 60.7%).

The analysis of the scatter plot of differences between the right and left sides demonstrates that the analyzed samples displayed a pairwise correlation of the PLS score between the right and right homologues landmarks that was (*r* = 0.942; *p* < 0.0001). PLS1 possessed 99.4% of total covariation score. This regression indirectly confirmed the absence of AS for shape, which was corroborated for size by a normal distribution of CS for each side (it would appear bimodal if AS existed) (Jarque-Bera *JB* = 3.91, *p* = 0.141) (Figure 4).

Finally, males presented a higher degree of asymmetry than females (*p* = 0.0282; *p*-value from permutation tests) for Mahalanobis distances among groups with 10,000 permutation rounds).

## 4. Discussion

A turtle’s shell probably reflects a trade-off between its protective function and an imposition on active dynamics [26]. The current work describes the use of geometric morphometrics in the study of curvature among four different *Testudo* species. This technique is based on multivariate space and size and was used with the collection of two-dimensional coordinates of paired definable outlines on the frontal view of the individual carapaces. The shapes of the lateral carapace curvatures in *Testudo* were computed and both sides compared. 

Perfect symmetry is hardly exhibited in live organisms. The presence of both fluctuating asymmetry (FA) and directional asymmetry (DA) was an unanticipated result but according to a seminal work [27], multiple asymmetries can exist at the same time. FA is defined as the small, completely random departures from bilateral symmetry [20], but its detected levels, although significative, were much lower than for DA.

DA, which is mostly conditioned genetically, occurs when there is a tendency for one side of a bilaterally present trait to be larger than the other [22]. As it is genetically regulated and therefore likely to have adaptive significance, this asymmetry must be seen as a result from functional optimization and not an imperfection. Carapaces could be shaped by the trade-off between self-righting and stability. The side effect—carapaces showing flared marginal scutes towards right side—remained statistically significant, thus confirming a side-bias on carapace asymmetrical shape. Moreover, as carapace asymmetry did not increase with size, it is assumed that turtle shells do not become increasingly asymmetrical with age.

Turtles are subjected to a variety of potentially destabilizing forces that can be either self-generated (e.g., propulsor movements) or external [28]. At this point, a possible speculative biological explanation for the detected asymmetrical pattern in *Testudo* carapace is that this asymmetry makes turning difficult (“static stability”, understood as the ability to resist passively turning of the body produced by destabilizing forces that damp stability forces), rather than having an effect on the easiness of self-righting potential (“kinematic instability”). For the sexual difference, one possible biological explanation explained in part to an increased efficiency of locomotion in males, which might allow them to move faster and with greater agility than females. Thus, a higher degree of asymmetry among males positively affects their self-righting. However, females can also lose their balance and fall on their back and moreover, females’ bodies are larger and thus, their agility is reduced, so DA is also present among them. In any case, it is important to account for sexual dimorphism in future asymmetry studies for other *Testudo* species.

It is widely known that animals possess body forms of bilateral symmetry. The structure of the *Testudo* carapace’s asymmetrical shape variation recorded in this study is dominated by a clear right tendency of curvature, which is congruent with some direct observations, at least among *Testudo hermanni,* which highlights a preferentially side self-righting [29]. In addition, we think that if the stomach is full, there is a risk of gastric torsion if a complete turn is made towards the same sense of the initial rolling. There has been a reported risk of gastric torsions in herbivorous tortoises [30].

Another question with respect to asymmetry is why detected directional asymmetry occurs predominantly on one side of the body. The bias-curved carapace of *Testudo* can be attributed in part to the internal asymmetry; the body asymmetry in many vertebrates is well illustrated by the left shifted location of the heart and asymmetry in visceral organs [31]. In turtles, the alimentary canal shows internal asymmetry [32]. This seems to be attributable to elongation of the organ to enhance its digestive efficiency. Internal asymmetry is found also in the vascular system [33].

This is the first report in frontal carapace curvature among *Testudo* to detail morphological features from the point of view of assessing geometrically bilateral asymmetry. It will be a useful reference dataset that can be applied for geographic comparisons of carapace shape of terrestrial turtles. We recommended using GM analysis because it is inexpensive and could be used on living organisms, avoiding destroying individuals. Moreover, our data, and any further results obtained with other species, may provide new insights into the anatomical pattern of carapace and of its functional form.

## 5. Conclusion

In *Testudo* species, carapace asymmetry seems to be dominated by a clear right directionality. A possible biological explanation is that this asymmetry makes the stable ventral turning difficult (“static stability”, understood as the ability to resist passively turning of the body produced by destabilizing forces).

## Figures and Tables

**Figure 1 animals-12-00471-f001:**
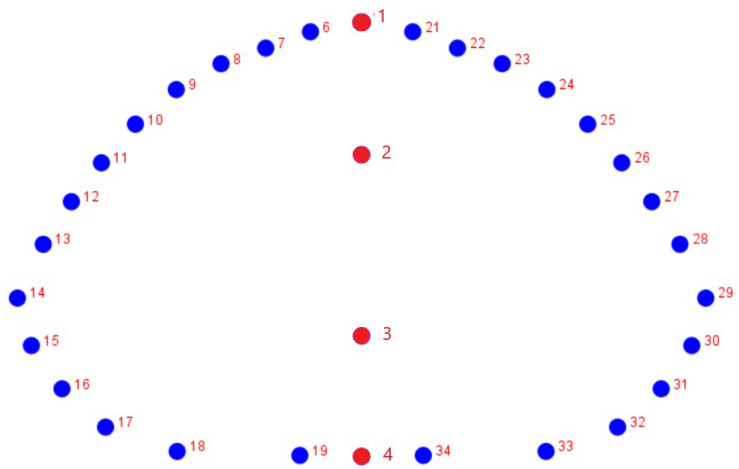
Landmarks configuration used in the study composed of 4 fixed landmarks (1 to 4) and two 15 semilandmark-curves imposed onto a 2D picture of *Testudo* sp. Frontal view.

**Figure 2 animals-12-00471-f002:**
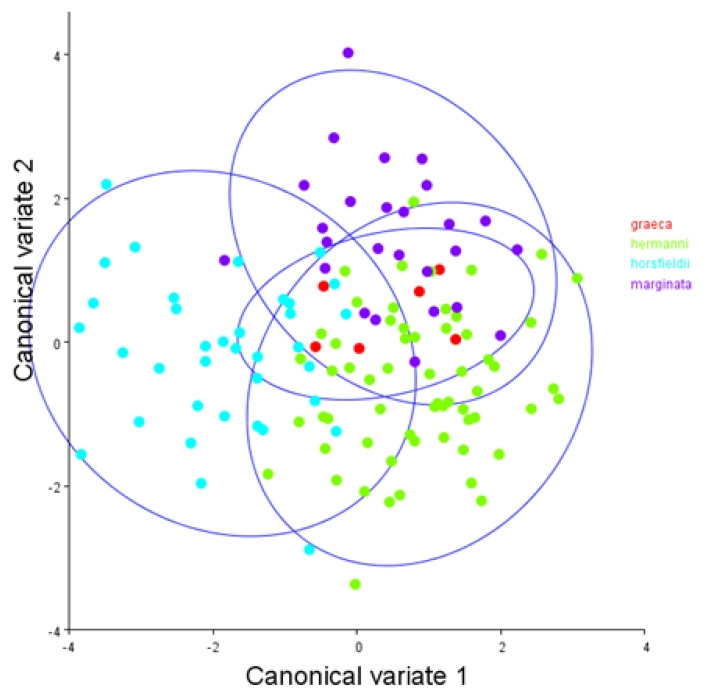
Canonical Variate Analysis for 4 *Testudo* species (*T. hermanni*
*n* = 30, *T. graeca*
*n* = 3, *T. marginata n* = 13 and *T. horsfieldii n* = 18) along the first two Canonical variate axes with 90% confidence ellipses for each species. This scatterplot shows an overlap among species.

**Figure 3 animals-12-00471-f003:**
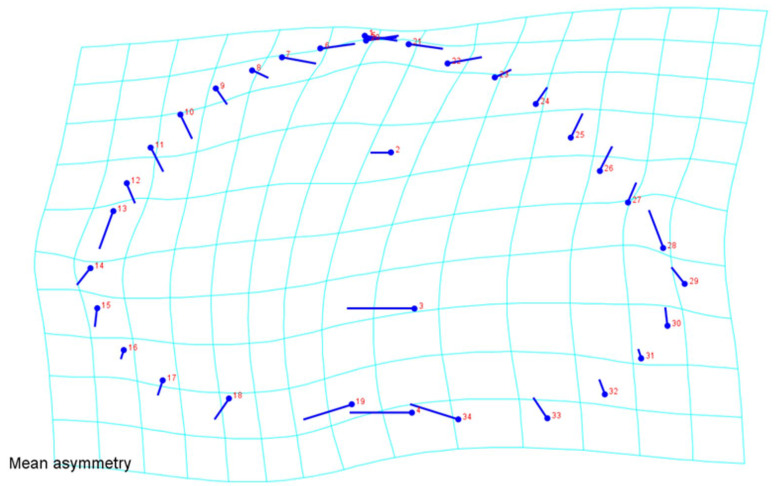
Deformation grid for the frontal view of *Testudo*. Directional asymmetry is shown as the difference between the averages of all original and reflected configurations. Transformation grids illustrate the shape changes from overall mean along Principal Component 1. Circles indicate the locations of the landmarks in the mean shape of the sample; sticks indicate the changes in the relative positions of the landmarks. It appears a clear shape patterns of lateral right displacement.

**Figure 4 animals-12-00471-f004:**
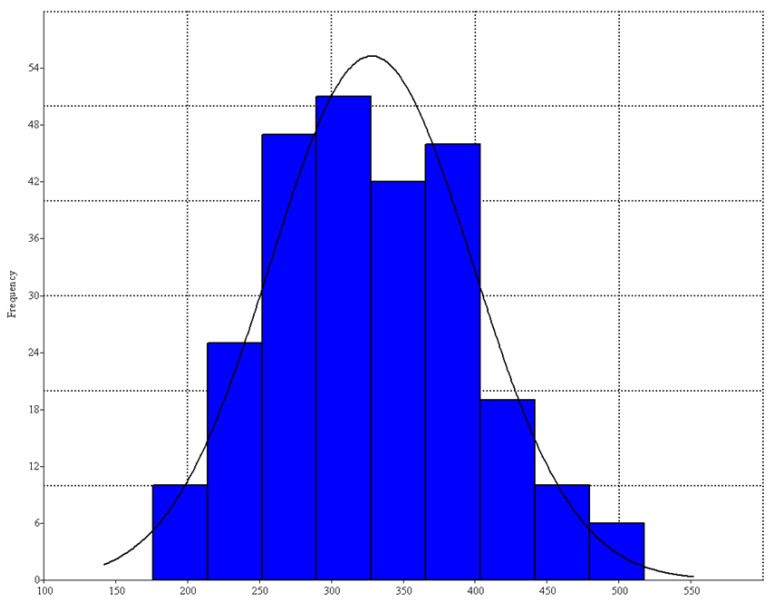
Distribution plot of centroid sizes for *Testudo* (*n* = 64) right and left parts of frontal carapace. The line shows the graph with a fitted normal distribution (*p* = 0.141). X-axis refers to centroid sizes.

**Table 1 animals-12-00471-t001:** Measurement error Procrustes ANOVA for size and shape of carapace symmetry for *Testudo n* = 64), with a significant effect of DA (Directional Asymmetry) for size and shape. Pillai trace was significative for DA (Pillai trace = 0.86; *p* < 0.0001) but also for FA (Fluctuating Asymmetry) (Pillai trace = 17.42; *p* < 0.0001). Sums of squares (SS) and mean squares (MS) are in units of Procrustes distances (dimensionless).

Effect	SS	MS	Degrees of Freedom	Fisher Test	*p*-Value
Size					
Individuals	1,784,840.5892940	28,330.8030050	63	47.17	<0.0001
Error	38,436.907440000	600.576679000	64		
Shape					
Individuals	0.451068730	0.000223744	2016	3.69	<0.0001
DA	0.013539950	0.000423124	32	6.99	<0.0001
FA	0.122113590	0.000060572	2016	0.32	1
Error	0.770446940	0.000188097	4096		

## Data Availability

The contents of all supporting data are the sole responsibility of the authors. The datasets generated and analyzed during the current study are available from the corresponding author upon reasonable request.

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
