# Peer review of "Symmetry-Breaking Stabilities in Carapace Curvature on Testudo (Reptilia, Testudinidae)"

_animals, 2022, doi:10.3390/ani12040471_

Round 1

Reviewer 1 Report

The authors present an interesting study on how morphological asymmetry could potential affect up-righting performance in tortoises. However, the presentation of the results (quality of figures and writing) needs much improvement.

Author Response

Quality images has been improved (higher resolution).

Reviewer 2 Report

The following article attempt to study the asymmetry variation in the turtle carpace, the methodology presented is interesting, nevertheless serius problems were found in the design of the study.

The most important problem was the way in which the authos test asymmetry with semilandmarks, please take a look the definition of a semilandmarks,  those slide between points, how do you pretend analyse variance between right and left side from those when they did not have vectorial movement?   this should be modified using the points as a landmarks itself.

I have several observation in the manuscript which were added to the pdf using comments function,  take a look and solve all ones  before consider to be accepted.  

The article idea is original and could be accepted, however it very important to the authors fix the quality of figures and also the comments in the pdf.

for now is accepted after major revisions

Author Response

All suggested changes have been introduced. Quality of images has been improved.

Reviewer 3 Report

The article describes an interesting aspect of the life of land turtles that is their capability to return to balance after overturning. Authors suggest that this event should be due to a little asimetry in the body form. The work sounds good as hypothesis and demonstration. Scientific approach is correct. Pheraps the only thing that could be improved are the figures, for example with a good photo of the 4 species of turtles (in dorsal, frontal, caudal and lateral views). These could help the readers to better understand the problem discussed. 

Author Response

Thanks for your comments.

I am going to ask to the Editor to incorporate pictures for each studied species.

Round 2

Reviewer 2 Report

I suggested changes in figure 1 as a schematic figure, that mean do not include a picture, see this example:  https://onlinelibrary.wiley.com/doi/10.1111/j.1463-6409.2008.00332.x

This reccomendation is fundamental due to is clearly a pixelated figure which is not for the quality of the journal, 

My acceptance of the article depend of the change of that figure for a draw, with landmarks, maybe can use and exportation of the same software maybe could be even better than the one is in the journal.

Author Response

Attached you will find the revised manuscript with a new schematic figure. 

We hope you will find it suitable to be published.

Thanks for your valuable comments.
